# Bilateral Permanent Childhood Hearing Loss and Health-Related Quality of Life in Adolescence

**DOI:** 10.3390/children8060484

**Published:** 2021-06-07

**Authors:** Stavros Petrou, Kamran Khan, Colin Kennedy

**Affiliations:** 1Nuffield Department of Primary Care Health Sciences, University of Oxford, Oxford OX2 6GG, UK; 2Warwick Clinical Trials Unit, University of Warwick, Coventry CV4 7AL, UK; k.a.khan@warwick.ac.uk; 3Faculty of Medicine, University of Southampton, Southampton SO17 1BJ, UK; crk1@soton.ac.uk; 4University Hospital Southampton NHS Foundation Trust, Southampton SO16 6YD, UK

**Keywords:** permanent childhood hearing loss, adolescence, health-related quality of life, preferences, health utilities index

## Abstract

Little is known about the impact of bilateral permanent childhood hearing loss (PCHL) on health-related quality of life (HRQoL). The objective of this study was to describe preference-based and non-preference based HRQoL outcomes in adolescence, from both self and proxy perspectives, amongst participants of the Hearing Outcomes Project. The Health Utilities Index Marks II (HUI2) and III (HUI3) and the PedsQL^TM^ Version 4.0 Generic Core Scales were used to measure HRQoL based on self and parent proxy reports in 114 adolescents aged 13–19 years, 76 with bilateral PCHL and 38 with normal hearing, recruited from a population sample that was followed up from birth to adolescence. Descriptive statistics and multivariable analyses were used to estimate the relationship between severity of PCHL and HRQoL outcomes. PCHL was associated with decrements in mean multi-attribute utility score that varied between 0.078 and 0.148 for the HUI2 (*p* = 0.001) and between 0.205 and 0.315 for the HUI3 (*p* < 0.001), dependent upon the national tariff set applied and respondent group. Multivariable analyses revealed that, after controlling for clinical and sociodemographic covariates, mean HUI3 multi-attribute utility scores were significantly lower in adolescents with moderately severe, severe and profound hearing loss than in adolescents with normal hearing. Significant differences in physical functioning, social functioning, psychosocial functioning and total PedsQL^TM^ scores were only observed when assessments by parents were relied upon, but these dissipated in the multivariable analyses. Bilateral PCHL is associated with poorer HRQoL outcomes in adolescence. Further studies conducted are needed to understand the trajectory and underpinning mechanisms of HRQoL outcomes following PCHL.

## 1. Introduction

Permanent childhood hearing loss (PCHL) is the most common sensory impairment in the early years. Bilateral PCHL of moderately severe, severe or profound severity affects approximately 1 in 750 children and is present at birth in more than 80% of affected children [1]. A recent systematic review and meta-analysis of studies conducted in very highly developed countries estimated a prevalence of bilateral permanent childhood hearing loss (PCHL) ≥26 dB HL detected through universal newborn hearing screening (UNHS) of 1.1 per 1000 screened children (95% confidence interval (CI): 0.9, 1.3) [2]. The prevalence was estimated at 5.9 (95% CI: 3.8, 8.4) per 1000 screened children when the analysis was restricted to children admitted to a neonatal intensive care unit [2].

PCHL can result in delays in communication, cognition, language, reading and social and emotional development during childhood, which in turn can result in lower educational and employment levels through adolescence and adulthood [3]. Globally, the current burden of resulting disability is estimated to be 1.80 years lived with disability per 1000 of the global population aged 0–19 years, equivalent to 43% of the burden of all intellectual disability and 45% of the burden of all epilepsy over the same range of ages [4]. This highlights the need for interventions to address the health-related quality of life (HRQoL) of those with PCHL during the era of the Millennium Development Goals.

Relatively little is known about the impacts of PCHL on multi-dimensional HRQoL outcomes. A recent systematic review and meta-analysis revealed that pooled estimates of these impacts are constrained by variations in the measurement method, age, hearing level and type of hearing loss amongst participants in contributing studies [5]. An analysis that restricted pooled estimates to four studies that had used the Pediatric Quality of Life Inventory^TM^ (PedsQL^TM^) revealed statistically significant and clinically important decrements relative to those of children with hearing within the normal limits (hereafter normal hearing for brevity) in the PedsQL^TM^ scores of those with bilateral PCHL in the social functioning and school functioning sub-scales. Differences in the physical domain scores were statistically significant but not clinically important. We have previously reported the results of the Hearing Outcomes Project in England, which assessed HRQoL outcomes in children with bilateral PCHL of moderately severe or greater severity [6]. The study found that, compared to children with normal hearing, children with bilateral PCHL had significantly lower multi-attribute utility scores at 7 to 9 years of age, as measured by the Health Utilities Index Mark III (HUI3) questionnaire [6]. However, the study necessarily relied on assessments by the principal caregiver rather than the child. Empirical evidence of the concordance between child and parent ratings of childhood health status suggests that parents are able to accurately rate observable behaviours, such as physical functioning and physical symptoms, but are less successful at identifying social or emotional impairments [7], highlighting the need for both self and proxy reports where possible [8].

Multi-dimensional HRQoL measures provide a holistic approach to capturing the consequences of PCHL that may be of concern to the children and families affected. These measures can be divided between preference-based instruments that capture individuals’ preferences (or *utility*) for the health state experienced and non-preference-based measures that have scoring systems that are not preference-based. For economists and other social scientists, preference-based HRQoL measures move beyond the narrow biomedical model for evaluative research and importantly generate outputs that satisfy the requirements of decision-making bodies concerned with cost–utility comparisons [9,10,11,12]. The objective of this study was to describe the HRQoL outcomes in adolescence of participants of the Hearing Outcomes Project using both preference-based and non-preference-based measurement approaches from both self and proxy perspectives.

## 2. Materials and Methods

### 2.1. Study Background and Sample

The study sample was drawn from 156,733 children included in two cohorts born in eight districts of Southern England between 1992 and 1997. The Wessex cohort was born over a 36-month period in four districts that formed the population for the Wessex Trial, a quasi-experimental trial in which universal newborn hearing screening (UNHS) was or was not undertaken during alternating 4–6-month periods in two pairs of hospitals, with UNHS equipment and personnel moving back and forth between the paired hospitals. The Greater London birth cohort was born in two pairs of health districts in Greater London over a 60-month period between 1992 and 1997 [13]. Each pair of health districts included one of the only two districts in the UK offering UNHS at that time and an immediately neighbouring district [14]. The language, reading, behaviour, economic and HRQoL outcomes in children with bilateral PCHL > 40 dB NHL in these two birth cohorts, and in a normally hearing comparison group, was assessed in 183 children (120 with PCHL and 63 with normal hearing) at a mean (standard deviation (SD)) age of 7.9 (1.2) years [6,14,15,16,17]. The normally hearing comparison group had been identified using an algorithm developed by the investigators based on the dates of birth of the required number of children with normal hearing within each district and for each screening period. Further assessment of the sample was undertaken in the Hearing Outcomes at Teen Age project at a mean age of 16.9 (SD: 1.4) years [18,19,20,21,22,23,24].

The study was approved by the Southampton and SW Hampshire Research Ethics Committee. Written informed consent was obtained from the teenage participants and their principal caregivers. Full details on the tracing and recruitment procedures of the children with PCHL and children with normal hearing and the measures used to assess outcomes during adolescence are reported elsewhere [18,19,20,21,22,23,24].

### 2.2. Medical Measures

Severity of PCHL was categorised from the most recent audiological evaluation at audiology and cochlear implant clinics as moderately severe (40–69 dB HL), severe (70–94 dB HL) or profound (≥95 dB HL) according to four-frequency averaging of the pure-tone thresholds at 0.5, 1, 2 and 4 kHz in the better ear [18,19,20]. The presence of additional medical conditions was informed by data extracted from medical records and parent reports. These included cerebral palsy, visual impairment and learning disability with the latter determined by a Ravens Progressive Matrices score equivalent to a non-verbal IQ less than 70 [25].

### 2.3. Health-Related Quality of Life Measures

In conjunction with the literacy, language, emotional and behavioural assessments, undertaken at the teenager’s home or at their school according to their expressed preference, each teenager and their principal caregiver (in all cases a parent) was independently interviewed about the teenager’s HRQoL. The interviews were conducted by a trained researcher who was unaware of the teenager’s audiological history. The HRQoL measures were the HUI and the PedsQL^TM^ Version 4.0 Generic Core Scales.

The HUI is a family of preference-based, multi-attribute utility measures [26]. The unedited 15-item questionnaire for usual health assessment was applied, which was obtained from the HUI developers and covers both Mark II (HUI2) and HUI3 health status classification systems. Self or proxy versions of the questionnaire were administered dependent upon whether the respondent was a teenager or a parent. The HUI2 health status classification system covers seven attributes: sensation, mobility, emotion, cognition, self-care, pain and fertility (although fertility is not assessed using the current HUI questionnaires), each with three to five levels. The HUI3 health status classification system covers eight attributes: cognition, vision, hearing, speech, ambulation, dexterity, emotion and pain, each with five or six levels. Responses to both the HUI2 and HUI3 health status classification system were converted into single attribute and multiplicative multi-attribute utility scores using published Canadian utility functions [26,27,28]. The HUI3 classification system is now recommended by the developers because of its broad applicability in both clinical and general population health studies, improvements in a number of definitions and an increased orthogonality of its attributes for structural independence [29]. Given that UK-specific utility algorithms are currently unavailable for the HUI3 classification system, we applied the Canadian algorithms that reflect the preferences of 504 general population adults living in the city of Hamilton, Ontario, who had previously been asked to value selected health states using both visual analogue scaling and standard gamble techniques [28]. In addition, we converted responses to the HUI2 health status classification system into utility scores using the algorithms developed by McCabe and colleagues on the basis of the preferences of 198 members of the UK general population, which might be considered more relevant for UK policy purposes [30].

The PedsQL^TM^ Version 4.0 Generic Core Scales were specifically designed to measure the core dimensions of health, as delineated by the World Health Organisation, as well as role (school/day care) functioning [31]. The PedsQL™ Version 4.0 Generic Core Scales comprise parallel age-specific self and parent proxy reports. The 13–18 (adolescent) modules were applied in this study using self (first person) or proxy (third person) versions of the questionnaire, which contain 23 items covering physical functioning (8 items), emotional functioning (5 items), social functioning (5 items) and school functioning (5 items). A 5-point response scale was utilised across each item (0 = never a problem; 1 = almost never a problem; 2 = sometimes a problem; 3 = often a problem; 4 = almost always a problem). Items were reverse-scored and linearly transformed to a 0–100 scale (0 = 100, 1 = 75, 2 = 50, 3 = 25, 4 = 0) with higher scores indicating better HRQoL. For scale and total scores, the mean was computed as the sum across all items divided by the number of items answered, thereby accounting for missing data. A psychosocial health summary score was calculated as the mean score over the number of items answered across the emotional, social and school functioning scales. Differences in baseline characteristics between the children with PCHL and the children with normal hearing that were followed up in adolescence were tested using the Student t-test for continuous variables and the Pearson Chi-squared test for categorical variables. For each of the attributes of the HUI2 and HUI3, we compared the proportion of adolescents with suboptimal levels of function (defined as below level 1 function) using Fisher’s exact test for equality of proportions. Differences in single-attribute and multi-attribute utility scores for both the HUI2 and HUI3 and differences in PedsQL^TM^ scale, psychosocial and total scores, between the adolescents with PCHL and those with normal hearing, were estimated using two-sample t-tests for unequal variance. The analyses were replicated based on self or parent proxy responses. Finally, we performed separate multivariable regressions to explore the effects of severity of PCHL (none, moderately severe, severe or profound) on HUI2 or HUI3 multi-attribute utility scores or the PedsQL^TM^ total scores. The multivariable analyses were replicated based on self or parent proxy responses. Covariates included in the multivariable models were informed by previous theoretical or empirical models [6,14,18,19,20] and included age at assessment (years), gender (male (referent, female), mode of communication (spoken language only (referent), both spoken and signed language, signed language only, gestural communication only), English main language (no (referent), yes), maternal highest educational qualification (no qualifications or <5 ‘ordinary’ (O) levels UK qualifications achieved at 16 years (referent), ≥5 O or ‘advanced’ (A) level UK qualifications achieved at 18 years, higher educational degree or postgraduate qualification), social class of the head of household (higher occupation (referent), intermediate occupation, lower occupation, never worked or long-term unemployed), annual family income (<£10,000 (referent), £10,000–£20,000, £21,000–£30,000, £31,000–£40,000, £41,000–£50,000, >£50,000) and experience of other significant medical condition and disorders (no (referent), yes). We used the Ordinary Least Squares (OLS) estimator to perform the multivariable regressions. For the multivariable regressions for which the multi-attribute utility scores were dependent variables, we assessed the robustness of our results using a Tobit estimator that accounts for the censoring of the dependent variable, which has an upper value of 1.0 [32].

Statistical analyses were conducted by using Stata 16.0 (Stata Corp, College Station, TX, USA). *p*-values of 0.05 or less were considered statistically significant.

## 3. Results

Of the 183 participants (120 with PCHL and 63 with normal hearing) followed up at primary school age, 114 teenagers and their families (76 (63.3%) with PCHL and 38 (60.3%) with normal hearing) agreed to participate in this latter phase of the research. Attrition was largely attributable to participants not responding to invitations to participate in this phase of the study [19,20,21]. Of the 76 participants with PCHL, 33 (43.4%), 19 (25%) and 24 (31.6%) had moderately severe, severe and profound hearing losses, respectively (Table 1). There were no significant differences between the clinical and sociodemographic characteristics of these 76 participants and those lost to follow-up in the larger sample of 120 children who had been assessed at primary school age [19]. There were no significant differences between the clinical and sociodemographic characteristics of the 76 participants with PCHL and the 38 with normal hearing that were followed up in adolescence with the exception that a higher percentage of adolescents with PCHL had additional medical conditions.

Comparisons of the frequency and proportion of suboptimal levels of function between the teenagers with PCHL and those with normal hearing are shown in Appendix A
Table A1 for the seven attributes of the HUI2 and the eight attributes of the HUI3, as reported by both the teenagers and the parent proxies. In two of the eight HUI3 attributes (hearing, speech), significantly higher proportions of suboptimal levels of function among teenagers with bilateral PCHL (*p* < 0.05) were reported by both the teenagers and the parent proxies. However, in a further four attributes (vision, ambulation, dexterity, cognition), significantly higher proportions of suboptimal levels of function were reported by the parent proxies but not by the teenagers.

Appendix ATable A2 presents the single-attribute utility scores for the Canadian and UK versions of the HUI2 and the Canadian version of the HUI3, based on reports by both the teenagers and the parent proxies. For the HUI2, the single-attribute utility scores were significantly lower in adolescents with PCHL for the sensation and self-care attributes (*p* < 0.05), regardless of the national tariff set applied or respondent group (Appendix A
Table A2). This was also true for the parent proxy reports which, in addition, also generated significantly lower single-attribute utility scores for the mobility and cognition attributes. For the HUI3, the single-attribute utility scores based on both self and parent proxy reports were significantly lower in adolescents with PCHL for the hearing and speech attributes and, additionally, based on parent proxy reports only, for the ambulation, dexterity and cognition attributes (Appendix A
Table A2).

Table 2 presents descriptive statistics for the HUI multi-attribute utility scores by severity of hearing loss. Regardless of measure, the national tariff set applied or respondent group, the mean HUI multi-attribute utility scores were significantly lower in the adolescents with PCHL as a cohort than in the group with normal hearing (*p* < 0.05) (Table 2). The mean HUI3 multi-attribute utility score based on self reports was estimated at 0.668 for the adolescents with PCHL as a cohort, compared with 0.872 for the adolescents with normal hearing, a mean difference in utility score of 0.205 that was statistically significant (*p* < 0.001). The difference in utility scores widened to 0.315 (0.580 vs. 0.894; *p* < 0.001) when parent reports were relied upon (Table 2).

Table 3 presents descriptive statistics for the PedsQL^TM^ scale, psychosocial and total scores. Based on self reports, there were no significant differences in mean scores the PedsQL^TM^ scale, psychosocial and total scores between the adolescents with PCHL and those with normal hearing (Table 3). However, based on reports by the parent proxies, mean scores were significantly lower for physical functioning (81.08 vs. 90.80: *p* = 0.009), social functioning (76.74 vs. 88.11: *p* = 0.003), psychosocial functioning (73.17 vs. 81.35: *p* = 0.022) and for the total score (76.25 vs. 84.64: *p* = 0.009) amongst adolescents with PCHL (Table 3).

Finally, the OLS regressions revealed that after controlling for the pre-specified clinical and sociodemographic covariates, and based upon self-reports by the teenagers, the mean HUI3 multi-attribute utility score for adolescents with moderately severe, severe and profound hearing loss was 0.136, 0.243 and 0.294 less, respectively, than that for adolescents with normal hearing (*p* < 0.05; Table 4). Adolescents with profound PCHL also had significantly lower multi-attribute utility scores than adolescents with normal hearing when variants of the HUI2 were used. However, there were no significant differences in mean PedsQL^TM^ total scores when adolescents with moderately severe, severe and profound PCHL were compared to adolescents with normal hearing. Similar patterns were found when parent reports of health-related quality of life outcomes were used (Table 5). The results of the multivariable regressions remained robust to the application of alternative estimators.

## 4. Discussion

This study revealed that decrements in HRQoL associated with bilateral PCHL that were previously observed at primary school age [6] persist into adolescence. PCHL is associated with significantly increased proportions of suboptimal levels of function for between one and four of the seven attributes of the HUI2 and for between two and six of the eight attributes of the HUI3, dependent upon the respondent group. PCHL is also associated with significantly lower single-attribute utility scores for between two and four of the seven attributes of the HUI2, and for between two and five of the eight attributes of the HUI3, dependent upon the national tariff set applied and respondent group. Notably, the decrements in mean multi-attribute utility score that varied between 0.078 and 0.148 for the HUI2 and between 0.205 and 0.315 for the HUI3, dependent upon the national tariff set applied and respondent group, exceed the 0.030–0.075 range of minimally important differences in utility score postulated in the literature as clinically important for evaluative purposes [33].

Other than hearing itself, the HUI3 attributes that were most likely to be suboptimal in this population were speech and cognition. Half of our study population had been born during periods with UNHS, and this was associated with benefits to language and to reading but not to speech, both at primary school age [14,15,16] and during the teenage years [19,20,21]. The modernising of audiology services to optimise the early intervention that was made possible by UNHS was, however, at an early stage in this 1990s birth cohort, so greater benefits would be expected in a 2020s birth cohort. Nonetheless, in the study population reported here, language scores at age 8 years, after taking reading ability at age 8 years into account, made a significant contribution to the prediction of reading comprehension at age 17 [21], providing longitudinal evidence of the benefit of gains in early childhood language to subsequent academic progress in this sample

A less consistent pattern of results was observed when the PedsQL^TM^ Version 4.0 Generic Core Scales were used as the health-related quality of life measure. Significant differences in scores for physical functioning, social functioning, psychosocial functioning and overall health-related quality of life were only observed when assessments by parents were relied upon, but even these dissipated in the multivariable analyses. A study by Borton and colleagues revealed no significant differences in PedsQL^TM^ Version 4.0 Generic Core Scales scores between 6–17 year old children with unilateral hearing loss and those with normal hearing [34]. In contrast, a study of 5–18 year old children with hearing loss in Singapore found that children using hearing aids had significantly lower scores in all subscales of the PedsQL^TM^ Version 4.0 Generic Core Scales except physical functioning in comparison to normally hearing children [35], whilst a study of 13–18 year old children with sensorineural hearing loss in Massachusetts, United States, revealed significantly lower PedsQL^TM^ Version 4.0 Generic Core Scale scores for school functioning only in comparison to population norms [36]. However, comparative assessments of our results against those reported in the broader literature are constrained by differences in definitions and categorisations of hearing loss, as well as ages at assessment.

A recent psychometric evaluation of the HUI3 and PedsQL^TM^ Version 4.0 Generic Core Scales in Australian pre-school children with low language or congenital hearing loss found that the PedsQL^TM^ may not be as sensitive as the HUI3 in distinguishing HRQoL impairments in children with and without low language [37]. It is likely that the hearing and speech attributes of the HUI3 allow it to be particularly sensitive at detecting the specific effects of hearing and language losses on the daily lives of children and adolescents, while the PedsQL may be more sensitive to social aspects of HRQoL that are particularly salient in these adolescents [22,23,24]. The HUI3 and PedsQL^TM^ Version 4.0 Generic Core Scales are each underpinned by distinct conceptual models with distinct attribute sets and scoring algorithms and as such should be viewed as complementary tools for assessing the HRQoL of children or adolescents with PCHL.

Differences in self and parent proxy assessments of HRQoL outcomes were observed across measures. A recent systematic review of self-reported versus proxy-reported assessments in the derivation of childhood utility values for economic evaluation revealed that parent proxies tended to generate lower utility values than child self reports [8]. Inter-rater agreement between child self reports and parent proxy reports was found to be particularly poor for more subjective attributes such as cognition, emotion and pain [8]. In our study, we found that parent proxies tended to report lower levels of function both for objective attributes that rely on observable behaviours (e.g., mobility and self-care for the HUI2, ambulation and dexterity for the HUI3 and physical functioning for the PedsQL^TM^ Version 4.0 Generic Core Scales) and also for more subjective attributes that are less observable (e.g., cognition for the HUI2 and HUI3 and emotional functioning for the PedsQL^TM^ Version 4.0 Generic Core Scales). Clearly, HRQoL assessments obtained from children or adolescents with hearing losses, and from their corresponding parent proxies, should be viewed as complementary, rather than necessarily interchangeable. Both sets of assessments may need to be considered in clinical practice and policy settings, including those informing cost-effectiveness-based decision-making.

As expected in any population based on a sample of youth with bilateral PCHL, a substantial minority (26%) also had a medical condition that might adversely affect their HRQoL additional to their PCHL. This accounts, in large part, for the reported decrements in ambulation, dexterity and, especially, vision and will also have contributed to the decrement in cognition apparent, for example, in the HUI3 scores. The effect, in multivariable analyses, of the presence of an additional medical condition on HRQoL was smaller than the effect of any degree of PCHL and non-significant by self report, but substantial and larger than the effect of PCHL by parent-report.

This study was based on a longitudinal design and population-based sample drawn from defined geographic areas rather than clinic-based populations; consequently, selection biases are unlikely to represent a major problem. It also used validated preference-based and non-preference-based measures [26,27,28,29,31], applied from multiple perspectives, to obtain a holistic view of HRQoL outcomes with assessments performed by a trained researcher who was unaware of the participants’ audiological history. There are, however, caveats to the study findings, which should be borne in mind by readers. First, the annual attrition rate of 3% over the approximately 17 years following UNHS or 4% over the approximately nine years following assessment at primary school age may have limited the statistical power of the study to detect potentially important effects of PCHL on HRQoL outcomes. Nevertheless, this attrition rate compares favourably with that observed in other childhood and adolescent cohort studies [38]. Moreover, we did not identify significant differences between the characteristics of the study participants and those that had been lost to follow-up. Second, the adolescents with PCHL were slightly older, on average, than those with normal hearing at assessment. However, our multivariable analyses revealed no significant relationship between age at assessment and HRQoL life outcomes. Third, the underpinning preference values (or utilities) attached to health states within the HUI health status classification systems were derived from surveys of adults conducted 15 to 25 years ago. This was necessitated by an absence of national tariff sets derived from adolescent samples. The development of preference-based HRQoL measures has largely overlooked the normative question of whose values are most valid for informing clinical and resource allocation decisions in the paediatric context. Arguably, the application of adolescent-derived values could have led to a different pattern of results [39,40]. Fourth, the assessments of adolescent HRQoL reported here were performed during the 2010s. Although we are not aware of any enduring changes in clinical practice or broader environmental changes in recent years that would have significantly altered the pattern of results, a replication of the analyses in a more recently born cohort would clarify the generalisability of our results.

How might the results of our study be used? If confirmed by future research, the data generated by our study provide a basis for targeting services towards adolescents likely to experience unfavourable HRQoL outcomes following bilateral PCHL. In addition, the health utility data reported in this study can act as a significant new resource that can inform quality-adjusted life year (QALY) estimation within this clinical context. The use of health utility catalogues to inform QALY estimation and cost-effectiveness estimates for preventive and treatment interventions is now relatively common in other areas of health care [41,42]. It is hoped, therefore, that our data will act as inputs into modelling-based economic evaluations of audiological interventions that rely on secondary data sources.

Since the present study shows that the largest decrements of HRQoL associated with bilateral PCHL are associated with the presence of that hearing loss and with the decrements in speech and cognition associated with it, early intervention to correct and mitigate the effects of hearing loss, aimed in particular to be of benefit to speech and language at teen age, can reasonably be expected to improve HRQoL. Based on the reports of parents, targeting services at adolescents with not only PCHL but also another medical condition, appears justified by their associated poorer HRQoL. In children with mild and moderately severe PCHL, aiding of hearing is correlated with levels of speech and language ability at 3 and 5 years of age with longer duration of hearing aid use being most beneficial for children who had the best aided hearing [43]. In children with severe to profound PCHL, the language development of children receiving a cochlear implant before the age of 12 months is reported to be on a par with that of their hearing peers [44]. The 2019 guidelines of the Joint Committee on Infant Hearing for early hearing detection and intervention [3] are of good quality [45] and provide comprehensive guidance on early detection and early intervention for PCHL, including aiding and cochlear transplantation. The present study provides an explicit rationale for using these evidence-based interventions early to improve speech and language and thus HRQoL.

In conclusion, the results of this study reveal that bilateral PCHL is associated with poorer HRQoL outcomes in adolescence in comparison to normal hearing. It complements evidence from a sparse body of literature on this topic [5,6,34,35,36,37,46]. Further longitudinal studies conducted from multiple perspectives are needed to understand the trajectory and underpinning mechanisms of HRQoL outcomes following PCHL6.

## Figures and Tables

**Table 1 children-08-00484-t001:** Sociodemographic and clinical characteristics of study population.

	By Severity of PCHL			All PCHL	Normal Hearing
Characteristic	Moderately Severe (*n* = 33)	Severe (*n* = 19)	Profound (*n* = 24)	(*n* = 76)	(*n* = 38)
Age (years), mean (SD)	16.9 (1.4)	17.6 (1.4)	16.9 (1.6)	17.1 (1.5)	16.4 (1.2)
Gender, *n* (%)					
Male	16 (48.5)	10 (52.6)	13 (54.2)	39 (51.3)	25 (65.8)
Female	17 (51.5)	9 (47.4)	11 (45.8)	37 (48.7)	13 (34.2)
Mode of communication, *n* (%)					
Spoken language only	22 (66.7)	12 (63.2)	13 (54.2)	47 (61.8)	37 (97.3)
Both spoken and signed language	9 (27.3)	5 (26.3)	9 (37.5)	23 (30.3)	0 (-)
Signed language only	1 (3.0)	0 (-)	0 (-)	1 (1.3)	0 (-)
Gestural communication only	0 (-)	1 (5.3)	1 (4.2)	2 (2.6)	0 (-)
Missing	1 (3.0)	1 (5.3)	1 (4.2)	3 (4.0)	1 (3.5)
UNHS status, *n* (%)					
Born in periods without UNHS	14 (42.4)	10 (52.6)	15 (62.5)	39 (51.3)	-
Age PCHL confirmed, *n* (%)					
>9 completed months	17 (51.5)	12 (63.2)	12 (50.0)	41 (54.0)	-
English main language at home, *n* (%)	33 (100.0)	17 (89.5)	20 (83.3)	70 (92.1)	36 (94.7)
Maternal educational qualifications, ^1^ *n* (%)					
No-qualifications/O-levels	8 (24.2)	4 (21.1)	7 (29.2)	19 (25.0)	6 (15.8)
≥5 O or A-levels	17 (51.5)	11 (57.9)	10 (41.7)	38 (50.0)	14 (36.8)
Degree/postgraduate	8 (24.2)	4 (21.1)	7 (29.2)	19 (25.0)	18 (47.4)
Social class, ^2^ *n* (%)					
Higher occupations	15 (45.5)	10 (52.6)	11 (45.8)	36 (47.4)	26 (68.4)
Intermediate occupations	10 (30.3)	3 (15.8)	5 (20.8)	18 (23.7)	8 (21.1)
Lower occupations	4 (12.1)	0 (-)	5 (20.8)	9 (11.8)	3 (7.9)
Never worked or long-term unemployed	4 (12.1)	6 (31.6)	3 (12.5)	13 (17.1)	1 (2.6)
Family income £GBP, *n* (%)					
<10,000	4 (13.3)	2 (11.8)	0 (–)	6 (8.8)	0 (-)
10,000–20,000	6 (20.0)	2 (11.8)	7 (33.3)	15 (22.1)	4 (10.8)
21,000–30,000	2 (6.7)	4 (23.5)	2 (9.5)	8 (11.8)	7 (18.9)
31,000–40,000	7 (23.3)	2 (11.8)	4 (19.1)	13 (19.1)	4 (10.8)
41,000–50,000	3 (10.0)	2 (11.8)	1 (4.8)	6 (8.8)	5 (13.5)
>50,000	8 (26.7)	5 (29.4)	7 (33.3)	20 (29.4)	17 (46.0)
Additional medical conditions, ^3^ *n* (%)	10 (30.3)	4 (21.1)	6 (25.0)	20 (26.3)	0 (-)
Hearing aids, *n* (%)					
No aid	3 (9.1)	2 (10.5)	12 (50.0)	17 (22.4)	-
Unilateral aid	4 (12.1)	1 (5.3)	3 (12.5)	8 (10.5)	-
Bilateral aids	26 (78.8)	16 (84.2)	9 (37.5)	51 (67.1)	-
Cochlear implants, *n* (%)					
None	31 (96.9)	17 (94.4)	11 (47.8)	59 (80.8)	-
Unilateral implant	1 (3.1)	1 (5.6)	7 (30.4)	9 (12.3)	-
Bilateral implants	0 (–)	0 (–)	5 (21.7)	5 (6.9)	-

PCHL, permanent childhood hearing loss. ^1^ O-levels refers to ‘ordinary level’ UK qualifications achieved at 16 years. A-levels refers to ‘advanced level’ UK qualifications achieved at 18 years. ^2^ Classified according to UK National Census 2002. ^3^ These were severe visual impairment, cerebral palsy and mental retardation. UNHS denotes universal newborn hearing screening; PCHL denotes bilateral permanent childhood hearing loss >40 dB.

**Table 2 children-08-00484-t002:** HUI multi-attribute utility scores in adolescents with normal hearing and with PCHL.

Group	*n*	Mean (SD)	Mean Decrement from Adolescents with Normal Hearing	*p* ^1^
HUI2 (Canadian); self-reported				
Normal hearing	38	0.917 (0.113)	-	
Moderately severe PCHL	28	0.864 (0.143)	−0.053	0.114
Severe PCHL	17	0.836 (0.117)	−0.081	0.024
Profound PCHL	22	0.783 (0.120)	−0.134	<0.001
Total PCHL	67	0.830 (0.132)	−0.086	0.001
HUI2 (Canadian); parent reported				
Normal hearing	37	0.929 (0.081)	-	
Moderately severe PCHL	31	0.832 (0.177)	−0.097	0.007
Severe PCHL	18	0.750 (0.237)	−0.179	0.006
Profound PCHL	23	0.738 (0.207)	−0.191	<0.001
Total PCHL	72	0.781 (0.205)	−0.148	<0.001
HUI2 (UK); self-reported				
Normal hearing	38	0.892 (0.121)	-	
Moderately severe PCHL	28	0.829 (0.110)	−0.063	0.031
Severe PCHL	17	0.815 (0.100)	−0.077	0.018
Profound PCHL	22	0.794 (0.078)	−0.098	<0.001
Total PCHL	67	0.814 (0.098)	−0.078	0.001
HUI2 (UK); parent reported				
Normal hearing	37	0.895 (0.112)	-	
Moderately severe PCHL	31	0.802 (0.160)	−0.093	0.009
Severe PCHL	18	0.731 (0.202)	−0.165	0.004
Profound PCHL	23	0.732 (0.176)	−0.163	0.004
Total PCHL	72	0.762 (0.177)	−0.133	<0.001
HUI3 (Canadian); self-reported				
Normal hearing	38	0.872 (0.163)	-	
Moderately severe PCHL	28	0.758 (0.185)	−0.115	0.012
Severe PCHL	17	0.646 (0.241)	−0.226	0.002
Profound PCHL	22	0.569 (0.201)	−0.303	<0.001
Total PCHL	67	0.668 (0.218)	−0.205	<0.001
HUI3 (Canadian); parent reported				
Normal hearing	37	0.894 (0.150)	-	
Moderately severe PCHL	31	0.655 (0.313)	−0.239	<0.001
Severe PCHL	18	0.534 (0.339)	−0.360	<0.001
Profound PCHL	23	0.514 (0.318)	−0.381	<0.001
Total PCHL	72	0.580 (0.324)	−0.315	<0.001

HUI, Health Utilities Index; PCHL, permanent childhood hearing loss; ^1^ Calculated using two-sample *t*-test for unequal variance.

**Table 3 children-08-00484-t003:** PedsQL Generic Core Scales health-related quality of life scale scores by severity of hearing loss.

Scale	Mean (SD) Scores		Mean Score Difference (95% CI)	*p* ^1^
	All PCHL (*n* = 76)	Normal Hearing (*n* = 38)		
Self-reported				
Physical functioning	85.59 (11.32)	85.86 (13.99)	−0.27 (−5.58 to 5.04)	0.920
Emotional functioning	80.07 (17.11)	76.71 (16.45)	3.36 (−3.38 to 10.11)	0.324
Social functioning	87.09 (12.77)	88.42 (15.03)	−1.33 (−7.11 to 4.45)	0.647
School functioning	76.72 (17.40)	77.11 (22.05)	−0.39 (−8.70 to 7.92)	0.926
Psychosocial functioning	81.29 (12.35)	80.75 (14.22)	0.55 (−4.95 to 6.05)	0.843
Total	82.79 (10.59)	82.46 (13.02)	0.32 (−4.62 to 5.27)	0.897
Parent reported				
Physical functioning	81.08 (25.65)	90.80 (12.45)	−9.72 (−16.92 to –2.51)	0.009
Emotional functioning	72.29 (21.25)	77.43 (20.16)	−5.14 (−13.41 to 3.13)	0.220
Social functioning	76.74 (21.53)	88.11 (16.26)	−11.37 (−18.69 to –4.05)	0.003
School functioning	70.49 (20.75)	78.51 (21.04)	−8.03 (−16.47 to 0.42)	0.062
Psychosocial functioning	73.17 (18.47)	81.35 (16.66)	−8.18 (−15.14 to –1.22)	0.022
Total	76.25 (19.08)	84.64 (13.46)	−8.38 (−14.65 to –2.12)	0.009

PedsQL, Pediatric Quality of Life Inventory; SD, standard deviation; PCHL, permanent childhood hearing loss; CI, confidence interval.^1^ Calculated by using two-sample t-test for unequal variances comparing all adolescents with hearing loss with adolescents with normal hearing.

**Table 4 children-08-00484-t004:** Multivariable analyses of self-reported HRQoL outcomes in relation to severity of hearing loss (OLS regression, fully specified).

Parameter	HUI3	*p*	PedsQL (Total Score)	*p*
	Adjusted Mean Difference (95% CI)		Adjusted Mean Difference (95% CI)	
Severity of hearing loss				
None ^1^				
Moderately severe	0.136 (−0.246 to –0.027)	0.015	0.019 (−7.069 to 7.106)	0.996
Severe	−0.243 (−0.375 to−0.111)	<0.001	−0.149 (−8.698 to 8.400)	0.972
Profound	−0.294 (−0.416 to−0.172)	<0.001	−2.176 (−10.079 to 5.726)	0.585
Age (years)	0.018 (−0.012 to 0.048)	0.238	1.194 (−0.766 to 3.154)	0.229
Female	0.035 (−0.046 to 0.116)	0.395	−1.813 (−7.05 to 3.427)	0.493
Mode of communication				
Spoken language only ^1^				
Spoken and signed language	−0.020 (−0.146 to 0.107)	0.758	3.835 (−4.339 to 12.010)	0.353
Signed language only	±		±	
Gestural communication only	±		±	
English main language				
No ^1^				
Yes	0.157 (−0.059 to 0.373)	0.153	4.048 (−9.930 to 18.027)	0.566
Maternal educational quals ^2^				
No quals/O-levels^1^				
≥5 O or A-levels	−0.058 (−0.179 to 0.063)	0.341	−4.150 (−11.942 to 3.642)	0.292
Degree/postgraduate	−0.089 (−0.221 to 0.043)	0.185	−6.257 (−14.808 to 2.293)	0.149
Social class ^3^				
Higher occupations ^1^				
Intermediate occupations	0.051 (−0.050 to 0.151)	0.319	0.275 (−6.223 to 6.773)	0.933
Lower occupations	0.038 (−0.121 to 0.196)	0.635	−1.389 (−11.636 to 8.857)	0.788
Never worked/unemployed	−0.039 (−0.467 to 0.388)	0.855	17.113 (−10.528 to 44.753)	0.221
Family income £GBP				
<10,000 ^1^				
10,000–20,000	0.001 (−0.455 to 0.457)	0.998	19.345 (−10.134 to 48.825)	0.195
21,000–30,000	−0.007 (−0.480 to 0.467)	0.978	20.379 (−10.238 to 50.997)	0.189
31,000–40,000	0.120 (−0.349 to 0.589)	0.611	21.947 (−8.359 to 52.252)	0.153
41,000–50,000	0.053 (−0.421 to 0.526)	0.826	27.072 (−3.515 to 57.659)	0.082
>50,000	0.065 (−0.407 to 0.537)	0.785	22.852 (−7.669 to 53.373)	0.140
Additional medical conditions ^4^				
No^1^				
Yes	−0.081 (−0.210 to 0.048)	0.217	−6.385 (−14.738 to 1.969)	0.132
Constant	0.420 (−0.182 to 1.023)	0.169	42.127 (3.154 to 81.099)	0.034
Adjusted R Squared	0.3013	-	0.0777	-

^1^ Reference category; ^2^ O-level refers to ‘ordinary level’ UK qualifications achieved at 16 years. A-levels refers to ‘advanced level’ UK qualifications achieved at 18 years; ^3^ Classified according to UK National Census 2002; ^4^ These were severe visual impairment, cerebral palsy, mental retardation and genetic syndrome.

**Table 5 children-08-00484-t005:** Multivariable analyses of parent-reported HRQoL outcomes in relation to severity of hearing loss (OLS regression, fully specified).

Parameter	HUI3	*p*	PedsQL (Total Score)	*p*
	Adjusted Mean Difference (95% CI)		Adjusted Mean Difference (95% CI)	
Severity of PCHL				
None ^1^				
Moderately severe	−0.124 (−0.236 to−0.013)	0.029	−2.713 (−11.128 to 5.703	0.523
Severe	−0.236 (−0.368 to−0.103)	0.001	0.661 (−9.342 to 10.663)	0.896
Profound	−0.159 (−0.280 to−0.038)	0.011	0.303 (−8.865 to 9.471)	0.948
Age (years)	0.008 (−0.023 to 0.038)	0.624	0.503 (−1.816 to 2.822)	0.667
Female	−0.047 (−0.129 to 0.035)	0.258	−3.206 (−9.396 to 2.983)	0.306
Mode of communication				
Spoken language only ^1^				
Spoken and signed language	−0.267 (−0.386 to−0.148)	<0.001	−10.527 (−19.503 to−1.550)	0.022
Signed language only	−0.163 (−0.625 to 0.300)	0.486	7.324 (−27.673 to 42.321)	0.678
Gestural communication only	−0.744 (−1.079 to−0.410)	<0.001	−40.467 (−65.797 to−15.137)	0.002
English main language				
No ^1^				
Yes	0.240 (0.025 to 0.455)	0.029	−9.000 (−25.278 to 7.280)	0.275
Maternal educational quals ^2^				
No quals/O-levels ^1^				
≥5 O or A-levels	0.087 (−0.028 to 0.202)	0.137	5.732 (−2.990 to 14.455)	0.195
Degree/postgraduate	0.044 (−0.087 to 0.176)	0.503	2.831 (−7.105 to 12.766)	0.572
Social class ^3^				
Higher occupations ^1^				
Intermediate occupations	0.072 (−0.029 to 0.174)	0.161	0.323 (−7.376 to 8.021)	0.934
Lower occupations	−0.045 (−0.202 to 0.112)	0.569	−1.441 (−13.318 to 10.435)	0.810
Never worked/unemployed	0.109 (−0.134 to 0.352)	0.376	1.577 (−16.817 to 19.972)	0.865
Family income £GBP				
<10,000 ^1^				
10,000–20,000	0.006 (−0.267 to 0.278)	0.967	−23.935 (−44.561 to−3.309)	0.023
21,000–30,000	−0.027 (−0.308 to 0.253)	0.847	−11.668 (−32.890 to 9.553)	0.277
31,000–40,000	−0.079 (−0.357 to 0.199)	0.573	−16.614 (−37.639 to 4.410)	0.120
41,000–50,000	0.040 (−0.248 to 0.328)	0.783	−17.670 (−39.482 to 4.142)	0.111
>50,000	−0.081 (−0.362 to 0.199)	0.566	−16.922 (−38.134 to 4.290)	0.116
Additional medical conditions ^4^				
No^1^				
Yes	−0.305 (−0.421 to−0.189)	<0.001	−17.667 (−26.429 to−8.905)	<0.001
Constant	0.547 (0.004 to 1.089)	0.048	99.808 (58.757 to 140.860)	<0.001
Adjusted R Squared	0.6144	-	0.3332	-

^1^ Reference category; ^2^ O-level refers to ‘ordinary level’ UK qualifications achieved at 16 years. A-levels refers to ‘advanced level’ UK qualifications achieved at 18 years; ^3^ Classified according to UK National Census 2002; ^4^ These were severe visual impairment, cerebral palsy, mental retardation and genetic syndrome.

## Data Availability

S.P., K.K. and C.K. had full access to all the data in the study and take responsibility for the integrity of the data and the accuracy of the data analysis. The authors are willing to share all unpublished data from the study with bona fide researchers who provide a methodologically sound proposal for use in achieving the goals of the approved proposal. The database can be made available to them through discussion with the corresponding author (S.P.), the chief investigator (C.R.K.) and the Wellcome Trust.

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
