# Peer review of "Bilateral Permanent Childhood Hearing Loss and Health-Related Quality of Life in Adolescence"

_children, 2021, doi:10.3390/children8060484_

Round 1
Reviewer 1 Report
This paper has a good concept and it is clear the authors spent many years collecting the data. I commend them for that.
Do not use the term “normal hearing” use youth who are hearing
The study title says that this study was on adolescents. The birth year of these participants is 1992-1997. These participants are now 24-29 years old. Was this study conducted 7-8 years ago? Would the results still be relevant when this study is published after more than 9 years? Please address this in the limitations.
It is not clear if the QOL questionnaires were valid and reliable. If they are please explicitly say that. If is it not please add to the limitations.
Discussion line 322- it says “that the” twice
These results are grim alluding to the fact that adolescents who are HH or Deaf have a lower QOL than their same age peers. The discussion is very shallow and does not point to any specific findings that can be alleviated by educational support, community services, or health interventions. Without this information and some concrete ideas, I believe this paper does not add as much to the literature as it illuminates a problem with no ideas for a solution. If the authors really want to make this paper impactful I would add an “Implications for Practice” section.
With some attention to the above issues this can be a nice addition to the existing literature.
Author Response
Reviewer 1
Do not use the term “normal hearing” use youth who are hearing
We would like to thank the reviewer for this suggestion. In response, we have amended the text on lines 26-27 (page 5 of clean version) to read as follows; “…children with hearing within the normal limits (hereafter normal hearing for brevity)”. This terminology follows the conventional nomenclature in the background clinical literature. The term ‘youth who are hearing’ can be misinterpreted to encompass children with degrees of hearing loss.
The study title says that this study was on adolescents. The birth year of these participants is 1992-1997. These participants are now 24-29 years old. Was this study conducted 7-8 years ago? Would the results still be relevant when this study is published after more than 9 years? Please address this in the limitations.
In response to the reviewer’s comment, we have added the following text to the limitations section of the discussion (lines 320-324 on page 16 of clean version): “Fourth, the assessments of adolescent HRQoL reported here were performed during the 2010s. Although we are not aware of any enduring changes in clinical practice or broader environmental changes in recent years that would have significantly altered the pattern of results, a replication of the analyses in a more recently born cohort would clarify the generalisability of our results.”
It is not clear if the QOL questionnaires were valid and reliable. If they are please explicitly say that. If is it not please add to the limitations.
In response to the reviewer’s comment, we have now clarified on pages 7-8 and on pages 15-16 that the Health Utilities Index Marks II (HUI2) and III (HUI3) and the PedsQLTM Version 4.0 Generic Core Scales are validated and reliable measures of childhood health-related quality of life, citing references 27, 28, 29, 30 and 32.
Discussion line 322- it says “that the” twice
We would like to thank the reviewer for this observation. This typo has now been corrected.
These results are grim alluding to the fact that adolescents who are HH or Deaf have a lower QOL than their same age peers. The discussion is very shallow and does not point to any specific findings that can be alleviated by educational support, community services, or health interventions. Without this information and some concrete ideas, I believe this paper does not add as much to the literature as it illuminates a problem with no ideas for a solution. If the authors really want to make this paper impactful I would add an “Implications for Practice” section.
In response to the reviewer’s comment, we have inserted a lengthy section into the discussion (pages 17-18 of clean version) describing the implication for practice of our research. The section reads as follows:
“How might the results of our study be used? If confirmed by future research, the data generated by our study provide a basis for targeting services towards adolescents likely to experience unfavourable HRQoL outcomes following bilateral PCHL. In addition, the health utility data reported in this study can act as a significant new resource that can inform quality-adjusted life year (QALY) estimation within this clinical context. The use of health utility catalogues to inform QALY estimation and cost-effectiveness estimates for preventive and treatment interventions is now relatively common in other areas of health care [40,41]. It is hoped, therefore, that our data will act as inputs into modelling-based economic evaluations of audiological interventions that rely on secondary data sources.
Since the present study shows that the largest decrements of HRQoL associated with bilateral PCHL are associated with the presence of that hearing loss and with the decrements in speech and cognition associated with it, early intervention to correct and mitigate the effects of hearing loss, aimed in particular to be of benefit to speech and language at teen age, can reasonably be expected to improve HRQoL. Based on the reports of parents, targeting services at adolescents with not only PCHL but also another medical condition, appears justified by their associated poorer HRQoL. In children with mild and moderately severe PCHL, aiding of hearing is correlated with levels of speech and language ability at 3 and 5 years of age with longer duration of hearing aid use being most beneficial for children who had the best aided hearing [42]. In children with severe to profound PCHL, the language development of children receiving a cochlear implant before the age of 12 months is reported to be on a par with that of their hearing peers [43]. The 2019 guidelines of the Joint Committee on Infant Hearing for early hearing detection and intervention [3] are of good quality [44] and provide comprehensive guidance on early detection and early intervention for PCHL, including aiding and cochlear transplantation. The present study provides an explicit rationale for using these evidence-based interventions early to improve speech and language and thus HRQoL.”

Reviewer 2 Report
This article evaluates the impact of a bilateral permanent childhood hearing loss (PCHL) on the health-related quality of life (HRQoL) of adolescents with self and parent proxy reports. While the Health utility index (HUI 2 and 3) were lower for the participants with PCHL, no significant differences were found for the PedsQL in the multivariable analysis.
This is an interesting article, as little is known to date about the HRQoL of adolescents with PCHL. The introduction is well written and gives a good overview of the topic. The methods describe all the necessary information. I would prefer to use only the term HUI 2 and 3 and not switch between HUI and Mark (e.g., line 179; Table 2 = Mark, Table 4= HUI). Different HUI versions are presented in the results. In my opinion, the tables have too many numbers and are overwhelming for the reader. The quality of the paper would improve if the tables focused more on the main results, for example, since there are no differences by severity for the PedsQL and the effects in the parents reports disappear in the multivariable analysis, I would show only the numbers for all PCHL in Table 3. In Tables 4 and 5, I would suggest showing only the results of HUI 3 and PedsQL, since the two versions of HUI 2 do not provide any additional information.
In the results (lines 228-230) the p-value for vision for the adolescents is not <0.05. Therefore, the sentence should be: In two of the eight HUI3 attributes (hearing, speech).... However, in a further four attributes (vision, ambulation…). The same in line 303 in the discussion.
Table 5 presents the results for the multivariable analyses for all parent-reported HRQoL outcomes. The final chapter of results describes these outcomes as similar to those for adolescents. In addition to the results of the adolescents, other factors such as mode of communication, main language, and most importantly, additional medical conditions were found to be highly significant for the parents. As this could be a reason for the different results between adolescents and parents and the study population differs in this last point (26.3% additional medical conditions for the adolescents with PCHL!) this should be mentioned and discussed.
In my opinion, it is also interesting to mention in the discussion, that the adolescents only differ from normal-hearing peers in the variable hearing and speech (HUI3) or sensation (HUI2) and have no other problems. This is also supported by the results of the PedsQL (lines 326-327), as no differences in social aspects were found in this study.
A search in PubMed with “quality of life hearing loss adolescent” yielded at least five papers among the first 50 citations. Such papers should be included in the discussion.
Since the differences in the parental PedsQL were no longer present in the multivariable analyses, I would suggest either mentioning this point in the abstract or removing the PedsQL details there (lines 23-26).
In the discussion, lines 368-369 describe that the application of adolescents-derived values could have led to a different pattern of results. Perhaps it should also be mentioned here that the set used is from 20-year-old surveys.
In the literature the new Joint committee on Infant Hearing, Year 2019, instead of the 2007, position statement should be citied [3, line 46].
Author Response
Reviewer 2
This is an interesting article, as little is known to date about the HRQoL of adolescents with PCHL. The introduction is well written and gives a good overview of the topic. The methods describe all the necessary information. I would prefer to use only the term HUI 2 and 3 and not switch between HUI and Mark (e.g., line 179; Table 2 = Mark, Table 4= HUI). Different HUI versions are presented in the results. In my opinion, the tables have too many numbers and are overwhelming for the reader. The quality of the paper would improve if the tables focused more on the main results, for example, since there are no differences by severity for the PedsQL and the effects in the parents reports disappear in the multivariable analysis, I would show only the numbers for all PCHL in Table 3. In Tables 4 and 5, I would suggest showing only the results of HUI 3 and PedsQL, since the two versions of HUI 2 do not provide any additional information.
We would like to thank the reviewer for their positive comments on the paper. We have now reduced the volume of information in the tables as recommended by the reviewer. We now show only the numbers for the entire PCHL group in Table 3 and now show only the HUI3 and PedsQL results in Tables 4 and 5.
In the results (lines 228-230) the p-value for vision for the adolescents is not <0.05. Therefore, the sentence should be: In two of the eight HUI3 attributes (hearing, speech).... However, in a further four attributes (vision, ambulation…). The same in line 303 in the discussion.
We would like to thank the reviewer for this observation. We have now made the requested corrections in lines 187-190 on page 11 of the clean version of the paper and line 236 on page 13 of the clean version of the paper.
Table 5 presents the results for the multivariable analyses for all parent-reported HRQoL outcomes. The final chapter of results describes these outcomes as similar to those for adolescents. In addition to the results of the adolescents, other factors such as mode of communication, main language, and most importantly, additional medical conditions were found to be highly significant for the parents. As this could be a reason for the different results between adolescents and parents and the study population differs in this last point (26.3% additional medical conditions for the adolescents with PCHL!) this should be mentioned and discussed.
In response to the reviewer’s helpful suggestion, we have now added the following text to lines 290-297 on page 15 of the clean version of the paper:
“As expected in any population based on a sample of youth with bilateral PCHL, a substantial minority (26%) also had a medical condition that might adversely affect their HRQoL additional to their PCHL. This accounts, in large part, for the reported decrements in ambulation, dexterity and, especially, vision, and will also have contributed to the decrement in cognition apparent, for example, in the HUI3 scores. The effect, in multivariable analyses, of the presence of an additional medical condition on HRQoL was smaller than the effect of any degree of PCHL and non-significant by self-report, but substantial and larger than the effect of PCHL by parent-report.”
In my opinion, it is also interesting to mention in the discussion, that the adolescents only differ from normal-hearing peers in the variable hearing and speech (HUI3) or sensation (HUI2) and have no other problems. This is also supported by the results of the PedsQL (lines 326-327), as no differences in social aspects were found in this study.
In response to the reviewer’s helpful suggestion, we have now added the following text to lines 245-255 on pages 13-14 of the clean version of the paper:
“Other than hearing itself, the HUI3 attributes that were most likely to be suboptimal in this population were speech and cognition. Half of our study population had been born during periods with UNHS and this was associated with benefits to language and to reading but not to speech, both at primary school age [14,16] and during the teenage years [20,22]. The modernising of audiology services to optimise the early intervention that was made possible by UNHS was, however, at an early stage in this 1990s birth cohort so greater benefits would be expected in a 2020s birth cohort. Nonetheless, in the study population reported here, language scores at age 8 years, after taking reading ability at age 8 years into account, made a significant contribution to the prediction of reading comprehension at age 17 [22], providing longitudinal evidence of the benefit of gains in early childhood language to subsequent academic progress in this sample.”
A search in PubMed with “quality of life hearing loss adolescent” yielded at least five papers among the first 50 citations. Such papers should be included in the discussion.
We would like to thank the reviewer for this observation. We have now added further background information to the discussion (page 18 of clean version of paper) and key new reference 45, 46 and 47 to the bibliography, which complement references 5, 6, 35 and 46.
Since the differences in the parental PedsQL were no longer present in the multivariable analyses, I would suggest either mentioning this point in the abstract or removing the PedsQL details there (lines 23-26).
We were guided by the limited word count allowed for the abstract. We have however inserted the following sentence, which hopefully clarifies that the multivariable analyses did reveal significant differences in mean HUI3 multi-attribute utility scores: “Multivariable analyses revealed that, after controlling for clinical and sociodemographic covariates, mean HUI3 multi-attribute utility scores were significantly lower in adolescents with moderately severe, severe, and profound hearing loss than in adolescents with normal hearing.” We do believe that it is important to report the PedsQL results in the abstract of the paper.
In the discussion, lines 368-369 describe that the application of adolescents-derived values could have led to a different pattern of results. Perhaps it should also be mentioned here that the set used is from 20-year-old surveys.
The text on lines 313-315 on page 15 of the clean version of the paper has now been revised to read as follows:
“Third, the underpinning preference values (or utilities) attached to health states within the HUI health status classification systems were derived from surveys of adults conducted 15 to 25 years ago.”
In the literature the new Joint committee on Infant Hearing, Year 2019, instead of the 2007, position statement should be citied [3, line 46].
We would like to thank the reviewer for this observation. Reference 3 has been replaced with the following citation:
Joint Committee on Infant Hearing. Year 2019 position statement: principles and guidelines for early hearing detection and intervention programs. J. Early. Hear. Detect. Interv. 2019, 4(2), 1-44.

Round 2
Reviewer 2 Report
Thanks to the authors for revising the paper. The tables are now much clearer and thus better to understand.
I still find that the presentation of the results of the PedSQL in the abstract is misleading, because significant effects are described, which in reality do not exist, as they dissipate in the multivariate analysis.
As already mentioned in the previous review, the comparison to the RESULTS of other studies is very poor (only the used tests Health Utilities Index II and III and the PedsQ have been discussed in detail). Therefore, such studies on quality of life in adolescents should not only be newly listed in the bibliography, but the results of these studies should also be compared and discussed with those described in the article.
Author Response
The following revisions have been made to the manuscript in the light of the reviewers’ comments:
Reviewer 2
I still find that the presentation of the results of the PedSQL in the abstract is misleading, because significant effects are described, which in reality do not exist, as they dissipate in the multivariate analysis.
In response to the comment, we have amended the results section of the abstract to read as follows:
“PCHL was associated with decrements in mean multi-attribute utility score that varied between 0.078 and 0.l48 for the HUI2 (P=0.001) and between 0.205 and 0.315 for the HUI3 (P< 0.001), dependent upon the national tariff set applied and respondent group. Multivariable analyses revealed that, after controlling for clinical and sociodemographic covariates, mean HUI3 multi-attribute utility scores were significantly lower in adolescents with moderately severe, severe, and profound hearing loss than in adolescents with normal hearing. Significant differences in physical functioning, social functioning, psychosocial functioning and total PedsQLTM scores were only observed when assessments by parents were relied upon, but these dissipated in the multivariable analyses.”
As already mentioned in the previous review, the comparison to the RESULTS of other studies is very poor (only the used tests Health Utilities Index II and III and the PedsQ have been discussed in detail). Therefore, such studies on quality of life in adolescents should not only be newly listed in the bibliography, but the results of these studies should also be compared and discussed with those described in the article.
In response to the comment, we have added the following section to the discussion and amended the reference ordering throughout accordingly:
“A study by Borton and colleagues revealed no significant differences in PedsQL Version 4.0 Generic Core Scales scores between 6-17 year old children with unilateral hearing loss and those with normal hearing [35]. In contrast, a study of 5-18 year old children with hearing loss in Singapore found that children using hearing aids had significantly lower scores in all subscales of the PedsQL Version 4.0 Generic Core Scales except physical functioning in comparison to normally hearing children [36], whilst a study of 13-18 year old children with sensorineural hearing loss in Massachusetts, United States, revealed significantly lower PedsQL Version 4.0 Generic Core Scale scores for school functioning only in comparison to population norms [37]. However, comparative assessments of our results against those reported in the broader literature are constrained by differences in definitions and categorisations of hearing loss, as well as ages at assessment.”